# Continuous Flow Chemistry: A Novel Technology for the Synthesis of Marine Drugs

**DOI:** 10.3390/md21070402

**Published:** 2023-07-13

**Authors:** Laura F. Peña, Paula González-Andrés, Lucía G. Parte, Raúl Escribano, Javier Guerra, Asunción Barbero, Enol López

**Affiliations:** Department of Organic Chemistry, Campus Miguel Delibes, University of Valladolid, 47011 Valladolid, Spain; laura_1196@hotmail.com (L.F.P.); paula.gonzalez.andres@uva.es (P.G.-A.); lucia.garcia.parte@estudiantes.uva.es (L.G.P.); raul.escribano@estudiantes.uva.es (R.E.); franciscojavier.guerra@uva.es (J.G.); asuncion.barbero@uva.es (A.B.)

**Keywords:** flow chemistry, photochemistry, marine drugs, novel technologies, drug discovery

## Abstract

In this perspective, we showcase the benefits of continuous flow chemistry and photochemistry and how these valuable tools have contributed to the synthesis of organic scaffolds from the marine environment. These technologies have not only facilitated previously described synthetic pathways, but also opened new opportunities in the preparation of novel organic molecules with remarkable pharmacological properties which can be used in drug discovery programs.

## 1. Introduction

Over the last years, novel technological approaches have revolutionized the way we understand science. Most of them have been based on automated techniques to accelerate productive processes, resulting in less human intervention and diminishing time. In addition, miniaturization techniques have also reduced waste production and subsequent treatment, and computational advances consider plenty of data to facilitate a more futuristic rational design. All these approaches and others are used to improve scientific projects in terms of quality, safety, and the environment, while, at the same time, improving people’s lives.

The emerging trend of adapting new technologies in chemistry has contributed to the development of new molecules that are not accessible otherwise. In this manner, inaccessible chemical space can be achieved to extend chemical diversity. Chemistry has traditionally been governed by flasks, in which almost any chemical reaction can be carried out. There was no need to adapt scientific equipment to the reaction outcome, as various reaction parameters (temperature, slow addition, concentrations, etc.) were usually modified to achieve the desired coupling reactions. However, continuous flow chemistry has largely been studied as an alternative over the past 20 years and allows chemists to adapt their equipment to the chemical methodology [1]. By using this technology, reagents flow in a continuous stream into a reactor where the chemical transformation takes place. The chance to carry out a reaction through small dimensioned tubes in an equipment with a higher surface-area-to-volume ratio presents several advantages such as the acceleration of reaction rates and an increase in terms of security. As shown below, this approach has several benefits for chemical manufacturing compared to batch processes and offers interesting advantages for the pharmaceutical industry, which has implemented this technology over the last years [2,3,4,5,6,7,8,9].

The benefits of flow versus batch processes are well-established [10,11]. Very likely, safety is one of the main concerns for the pharmaceutical industry. If hazardous reagents, or unstable or toxic intermediates participate in the reaction, they are not accumulated but are rapidly consumed in the small reactor size [12]. Similarly, the lower space of vapor pressure in the reactor decreases the risk of explosion, for instance, in hydrogenation or carboxylation reactions. The reaction rate in continuous flow can also be increased when high temperatures or pressures are installed. In addition, the better heat and mass transfer processes in flow allows us to scale up exothermic reactions, avoiding the risks of heat removal. There is also an exquisite control of the reaction parameters (e.g., residence times, concentrations, and temperatures) which is usually related to the obtention of more selective transformations [13]. Consequently, an increase in the reaction yield and purity are observed with a small footprint.

Continuous flow technology is also very versatile in terms of set-up (Figure 1). Based on the requirements of the reaction, several pieces (reactors, mixers, pumps, and tubing) can be installed to increase productivity. It is also possible to carry out an in situ analysis of the reaction outcome by installing analytical techniques (IR and NMR spectroscopy, and MS) or even subsequent purification systems (e.g., flash chromatography and HPLC). Furthermore, novel synthetic techniques such as photochemistry, electrochemistry, or artificial intelligence can be implemented, which overcome some limitations found in the batch mode [14]. Automated systems have also been reported in continuous flow chemistry. These results are usually related to an increase in process sustainability as well as a considerable reduction of both cost and time.

The chance to carry out organic synthetic transformations more safely, cleanly, and with shorter reaction times has attracted the attention of the pharmaceutical industry. Many Big Pharma organizations such as Eli Lilly, Janssen, Novartis, GSK, or Pfizer have implemented this technology in key steps of API manufacture. Although some of these methodologies have been reported in international journal articles (preparation of abemaciclib [15] and nevirapine [16]), most of them are covered by patents with few details of the chemical transformation [17]—for instance, the synthesis of brivaracetam [18], valacyclovir [19], or crizotinib as an intermediate [20]. Regarding drug discovery, the benefits of flow technology have also been considered but not significantly explored [21,22].

On the other hand, the ocean is still an underexplored environment which can be explored by scientists to identify new chemical analogs. Here, the chances of finding new molecules with interesting drug-like properties are remarkable due to the different evolution pathways developed by living organisms over the years. In this regard, several biologically active families of organic compounds have been found [23,24,25,26]. It is estimated that approximately 3000 entities have been identified from marine sources in which some of them have provided valuable information in the search of new therapies [27]. These sophisticated skeletons have created great interest in marine pharmacology which has been reflected in the development of new chemical methodologies to obtain these molecules synthetically. In addition to these great efforts, additional technological approaches based on flow chemistry and photochemistry have also been described during the last decade. In this review, we pretend to summarize how these technologies have facilitated the synthetical preparation of molecules from marine organisms and their precursors to illustrate the benefits of using these synthetic tools.

## 2. Synthesis of Marine Drugs Using Flow Chemistry

Böstrom, Brown, and coworkers summarized the most popular reactions in pharmaceutical companies, mentioning that amide formation, Suzuki–Miyaura cross-coupling, reductive amination, nucleophilic substitution, and Boc protection/deprotection steps are the most common [28]. Many of them involve carbon–heteroatom or carbon–carbon bond formations. The construction of organic scaffolds using these approaches is tremendously useful in the pharmaceutical industry, so many continuous flow methodologies have been reported for this type of transformations, highlighting the advantages of using flow instead of batch. In the case of marine drugs, most of the novel synthetic approaches developed using continuous flow and photochemistry have been based on this type of coupling reactions.

### 2.1. Synthesis of Aplysamines

Aplysamines are bromotyrosine-derived secondary metabolites that are isolated from different sponges [29,30,31]. Aplysamine 6 (**1**) was isolated in 2008 from the Australian marine sponge *Pseudoceratina* sp. by Quinn and coworkers [32]. It contains one bromotyrosine unit and one bromomethoxycinnamoyl unit (Figure 2) and shows the inhibition of isoprenylcysteine carboxyl methyltransferase (Icmt) with an IC_50_ value of 14 μM.

The first total synthesis was reported in 2009 by Ullah and Arafeh using a traditional batch approach [33]. The addition of the acid chloride (**3**) to a solution of dibrominated agent (**2**) occurred at 5–10 °C in a 1:1 mixture of THF/DMF and using 2,6-lutidine as base. This reaction generated amide (**4**) in a 70% yield after column purification. Alkylation of (**4**) with an excess of 1,3-dibromopropane (**5**) in DMF using potassium carbonate as the base generated (**6**) an excellent yield (95%). Then, treatment of (**6**) with an excess of ammonium hydroxide at 80 °C produced the final aplysamine 6 (**1**) with an overall yield of 27% (Figure 1).

Years later, in 2010, Organ and coworkers reported the synthesis of the same molecule using a continuous flow methodology [34]. The key step was a microwave-assisted continuous-flow organic synthesis (MACOS) Heck reaction. The synthesis started with the acylation of (**7**) using EtN(*i*Pr)_2_ in DMF to provide the intermediate (**9**). The crude of this reaction was directly used in the next step. The coupling of dibrominated agent (**9**) with aryl iodide (**10**) in a microwave-assisted Heck reaction produced (**11**) with a 53% yield. The next step was the alkylation of phenol (**10**) to provide (**13**) in a 96% yield. For phenol deprotonation, potassium trimethylsilanoxide (**12**) was loaded into a syringe with (**11**), while the alkyl bromide (**13**) was loaded into a second syringe to be pumped into a reaction loop. The final step was the deprotection of the compound (**14**) with TFA. As it could not be carried out in DMF, another step was necessary to change the solvent to CH_2_Cl_2_ and provide (**1**) with a 90% yield (Figure 2). With this synthetic route, aplysamine 6 was obtained with a 46% overall yield in four steps and in a final quantity of 0.63 g. Comparing the batch methodology with the continuous flow approach, the latter provides the product in fewer reaction steps, with higher yields, and it could easily been scaled up.

### 2.2. Synthesis of (−)-hennoxazole A

(−)-Hennoxazole A (**15**) is a natural marine product that comes from the sponge *Polyfibrospongia* from Miyako Island coast, Japan. It was first isolated in 1991 by Scheuer and coworkers at the University of Hawaii. Hennoxazole A is one of the most active antivirals against the herpes simplex virus type 1 of its group (IC_50_ = 0.6 µg/mL) and has peripheral analgesic activity comparable with indomethacin [35]. This unique feature is related to its structure, directly linked to its bisoxazole core, which is found in various marine compounds like alkaloid diazonamides and the hennoxazole family. Hennoxazole A (15) has a highly functionalized tetrahydropyranyl ring moiety and a non-conjugated triene unit (Figure 3) [36].

In 1995, Wipf and Lim synthesized (−)-hennoxazole A (**15**) for the first time in a synthesis which required 42 batch steps, with a minimal reaction yield [37]. However, it was not until 2013 that Fernández and coworkers developed a continuous flow approach in a synthesis which involves both batch and flow methodologies [38]. The synthetic strategy relies on the preparation of the three fragments (**18**), (**19**)**,** and (**20**) (Figure 3).

Fragments (**18**) and (**20**) were prepared in batch mode. However, for the synthesis of bisoxazole core (**19**), flow technologies were used to avoid laborious synthetic batch protocols using a Vapourtec R2+/R4 flow system (Figure 4). All reactions took place in a conventional flow coil reactor (CFC) with a volume of 10 mL. Therefore, a straightforward coupling between 5-pentenoic acid (**21**), activated in situ with carbonyldiimidazole (CDI), and (±)-serine methyl ester (**22**) in the presence of triethylamine took place, followed by successive cyclodehydration with diethylaminosulfur trifluoride (DAST) to form oxazoline (**23**). It was necessary to quench the excess of DAST and to capture the generated HF, improving the safety profile by using a CaCO_3_/SiO_2_ scavenger. Intermediate (**24**) was prepared, oxidizing intermediate (**23**) with bromochloroform in the presence of DBU. Then, derivate (**24**) was subjected to a saponification step with aqueous NaOH to give acid (**25**), which was converted to bisoxazole ester (**27**) using an oxazole-forming process. A batch reduction of (**27**) with DIBAL-H produced aldehyde (**19**) in good yield (93%). This flow process avoids aqueous extraction and column chromatography purification by using polymer-supported reagents and scavengers to purify intermediates. These techniques involve less time workup, and it also less time-consuming through a telescope synthetic protocol. Derivative (**27**) was also achieved on the multigram scale (<5 g).

Once the three fragments were prepared in the gram scale in both batch and flow methods, they were assembled to form the (−)-hennoxazole A (Figure 5). Ketone (**18**) reacted with aldehyde (**19**) using (−)-Ipc_2_BCl, followed by an in situ reduction to obtain diol (**28**) as a single diastereomer. The next step consisted of a gold-catalyzed cyclization to form the tetrahydropyran ring intermediate (**16**), which was then methylated to prepare (**29**). The addition of fragment (**17**) to intermediate (**29**) required a second-generation Grubbs complex to promote the formation of (**30**). For more steps, involving deprotection and olefination steps generated (−)-hennoxazole A (**15**) with a 19% overall yield.

Thus, the synthesis of (−)-hennoxazole (**15**) has been completed in 26 steps with a good overall yield and by using a combined batch/flow approach. Synthetic batch strategies required work-up and purification strategies, but the telescope synthesis promoted by flow reduced the time required by employing scavengers and polymer supports to purify intermediates along the synthetic process.

### 2.3. Synthesis of Vidarabine

1-β-D-arabinofuranosyladenine (**31**), most commonly known as vidarabine or ara-A (Figure 4), is a synthetic analog of spongosine, a nucleoside isolated from the Caribbean sponge *Cryptotethya crypta* in 1951 by Bergmann and Feeney [39]. Vidarabine is active against herpes viruses, poxviruses, and certain hepadnarviruses, rhabdoviruses, and RNA tumor viruses, and it also shows activity against the vaccinia virus in both in vitro and in vivo studies. It has been used for herpes encephalitis and other herpes infection treatments. However, nowadays, it is no longer used due to newer and more efficient antiviral drugs currently available in the market. Vidarabine was one of the three marine-derived drugs approved by the U.S Food and Drug Administration (FDA) and the first antiviral nucleoside to be licensed for the treatment of systematic herpes virus infection [40].

Synthetic nucleosides are often synthetized by multistep chemical processes, which leads to the formation of undesired by-products and low overall yields. Nucleoside phosphorylase would solve this problem, simplifying nucleoside synthesis through the transfer of a sugar residue to a second nucleobase, obtaining a transglycosylation process [41].

In this context, a flow-biocatalyzed synthesis of vidarabine using nucleoside phosphorylases was reported by Tamborini et al. [42]. Authors used an enzymatic twosome that had previously been used successfully for the synthesis of vidarabine in batch mode: a uridine phosphorylase from *Clostridium perfrigens* (*Cp*UP) and a purine nucleoside phosphorylase from *Aeromonas hydrophila* (*Ah*PNP). These two enzymes were co-immobilized on two hydrophilic supports, glyoxyl-agarose (GA) and EziG^TM1^ (Opal), by flowing a solution of the phosphorylases through them. The stability of both bioreactors was studied and the GA bioreactor was thereafter chosen because the conversion achieved with the EziG^TM^ bioreactor decreased to less than 10% due to the enzyme lost in the exiting flow stream. Thus, a bioreactor of 10 mL volume, 15 mm diameter, 150 mm length, and with 10 g of GA was prepared. A solution containing arabinofuranosyl uracil (**32**, araU, 16 mM) as the sugar donor and adenine (**33**, 8 mM) as the sugar acceptor in 50 mM phosphate buffer pH = 7.5 (1 L) was pumped through the bioreactor at 83 μL/min and pressurized at 20 psi, with a residence time of 120 min and a temperature of 28 °C, achieving a 67% conversion. The exiting flow stream was collected into a cooled vessel (4 °C), where vidarabine precipitated due to its poor water solubility. Finally, vidarabine was recovered by filtration under vacuum, washed with cold water, and dried. After 8 days of continuous work of the bioreactor, vidarabine (araA, **31**) was obtained with a 55% yield and >99% purity (Figure 6).

More studies about the vidarabine bioenzymatic flow synthesis were performed by Calleri et al. [43] In this case, two different immobilized enzyme reactors (IMERs) were prepared, immobilizing the enzymes on monolithic supports. On the one hand, they immobilized *Cp*UP and *Ah*PNP on an aminopropyl silica monolithic column functionalized with glutaraldehyde. This IMER was placed in a chromatographic system, after which the 5 mL reaction mixture containing the araU (10 mM) and adenine (5 mM) in 10 mM phosphate buffer (pH = 7.0) was continuously pumped through the bioreactor at 37 °C and with a flow rate of 0.5 mL/min. On the other hand, a monoenzymatic IMER was prepared, immobilizing the nucleoside 2′-deoxyribosyltransferase from *Lactobacillus reuteri* (*Lr*NDT) on an epoxy monolithic support. Same reaction conditions used in the *Cp*UP-*Ah*PNP IMER were applied. While the bienzymatic IMER reached a conversion of 60% in 24 h, the monoenzymatic IMER achieved a conversion of <2% in 24 h. Therefore, the *Cp*UP-*Ah*PNP IMER turned out to be more convenient for the vidarabine biocatalyzed flow synthesis.

The vidarabine flow synthesis showed some advantages over batch mode. This last synthesis was reported by Serra et al. [44], using *Cp*UP and *Ah*PNP as the co-immobilized catalysts, and achieved a 53% product yield and a 99% purity (Figure 7). Therefore, in terms of yield and purity, the flow-based approach does not show important advantages. On the other hand, the batch mode synthesis required the use of DMF as a co-solvent to maintain the enzyme activities and enhance product solubility, therefore preventing its precipitation and, thus, facilitating its separation from the immobilized enzymes. This places the flow approach over the batch mode in terms of green chemistry, since DMF is not the greenest solvent and has a significant environmental impact.

Finally, if the established chemical synthesis of vidarabine is taken into account, the advantages of the biocatalyzed flow-based approach are even clearer. This synthesis begins with the reaction between *N*-benzoyladenine (**35**) and 2,3,5-*O*-benzyl-D-arabinofuranosyl chloride (**34**) to give the product (**36**). The hydrogenation of this intermediate affords vidarabine (**31**), with an overall yield of 41% (Figure 8) [45].

By comparing the yields obtained with each approach, a clear improvement can be observed with the flow-based synthesis approach. On the other hand, if green chemistry principles are considered, the flow-based approach is preferable to the chemical synthesis one, since it reduces the solvents used, as well as the number of steps and reaction times required; the use of enzymes as catalysts also helps to reduce the energy requirements and make the process more efficient, as the bioreactor can operate for almost 8 days without losing activity.

### 2.4. Synthesis of Eribulin

Eribulin mesylate or Halaven^®^ (**37**, Figure 5) is a synthetic macrocyclic ketone derivative of the marine natural product halichondrin B, which was isolated from the marine sponge *Halichondria okadai* in 1986 [46]. Eribulin inhibits the growth phase of microtubules without affecting the shortening phase and sequesters tubulin into non-productive aggregates [47]. For this reason, in 2010, after Phase III trials, FDA approved eribulin for the treatment of metastatic breast cancer [48,49]. A few years later, eribulin mesylate (**37**) was also approved for patients with metastatic or unresectable liposarcoma who have received a prior ineffective treatment [50]. Nowadays, eribulin has been approved in over 50 countries.

Due to the biological importance and the commercial use of this drug, many synthetic routes have been developed [51,52,53,54]. However, some commercial synthetic problems are related to high operating costs, mainly because of the high energy requirements for reactor cooling and long operation times (Figure 9) [55]. For this reason, new technologies such as continuous flow have been applied to cryogenic reactions [56].

Reduction and coupling reactions, under DIBAL-H and *n*-BuLi conditions, respectively, were studied by K. Tagami and coworkers using a flow system Micro Process Server (MPS-α200) and the corresponding micromixers (CMPS-α02 and XMPS-γ101H), which could be used in a plant-scale operation [56].

The reduction of ester (**38**) to aldehyde (**39**) was optimized to reduce the formation of the over-reduced alcohol. Two different solutions, ester (**38**) in toluene and DIBAL-H (1 M in toluene), were fed into the first micromixer. A line delivering acetone in toluene was incorporated before the quenching with HCl 1 M. After running the flow system continuously for 87 min, (**39**) was almost quantitatively obtained (98.1%, HPLC area). The temperature was controlled by submerging the reactor and tubing in a cooling bath at −50 °C (Figure 10). Despite the requirement of cryogenic conditions in this methodology, the surface area of heat transfer was small and better energy efficiency was obtained [56].

Consecutively, the *n*-BuLi-mediated coupling reaction between aldehyde (**39**) and sulfone fragment (**40**) was optimized. For this process, two micromixers were positioned sequentially and solutions of sulfone (**40**) in THF, *n*-BuLi (1.6 M in hexane), and aldehyde (**39**) in heptane were fed into the system via syringe pumps. After mixing, the reaction was immediately quenched with an aqueous solution of NH_4_Cl. To obtain a better conversion, the flow rate was increased to 20 mL/min and the reaction was performed at 10 °C, a higher temperature than in batch conditions (Figure 11). To avoid the degradation of compound (**41**), residence times could not be longer than 2.4 s for lithiation and 2.1 s for the coupling reaction [56].

On the other hand, avoiding expensive metal catalysts is also a challenge in industrial synthesis. In this regard, a chemoenzymatic process has recently been developed for the preparation of optically pure alcohol (**45**), a precursor of the building block (**44**) in the preparation of the C14–C19 intermediate (**42**) of eribulin (Figure 12) [57].

The desired propargyl alcohol (*S*)-**45** was previously prepared by Cr(III)-mediated catalytic enantioselective propargylation of the corresponding aldehyde (**47**) with propargyl bromide (**46**), in a 73% yield and with 87% *e.e*. To obtain the pure enantiomer, the reaction was performed with the enzyme dissolved in an aqueous buffer (Figure 13) [58].

The recovery and recycling of the enzyme was difficult under these conditions (Scheme **13**). Thus, S. Gosh et al. proposed a continuous mode to synthesize the pure enantiomer (*S*)-**45** by enantioselective enzymatic acylation of the racemic homopropargyl alcohol (±)-**45**. Firstly, racemic homopropargyl alcohol (±)-**45** was obtained in a 69% yield from pent-4-en-1-ol (**48**) in three reactions steps: (i) protection of hydroxy group, (ii) ozonolysis, and (iii) propargylation. Then, a continuous kinetic resolution was carried out in a jacketed stainless-steel reactor at 27 °C with a circulating water bath. A mixture of the alcohol (±)-**45** and vinyl acetate dissolved in methyl tert-butyl ether (MTBE) was pumped into the packed-bed reactor (PBR). After the enzymatic reaction, optically pure homopropargyl alcohol (*R*)-**45** and acylated (*S*)-**49** were obtained. The enzyme packed in the reactor could be used continuously for 1 week, maintaining a productivity of 1.12 mM/h/g. Finally, both compounds were treated and isolated properly to produce the desired pure alcohol (*S*)-**45** with a 90% overall yield and >99% *e.e* (Figure 14) [57].

### 2.5. Synthesis of Yessotoxin

Yessotoxin (YTX, **50**) is a disulfated polyether marine toxin which was first isolated in 1987 from the digestive glands of the scallop *Patinopecten yessoensis* [59]. A few years later, Satake et al. demonstrated that the marine dinoflagellate *Protoceratium reticulatum* was responsible for the biogenetic origin of yessotoxin (**50**) [60]. Its structure and absolute stereochemistry was elucidated by NMR spectroscopy using different techniques (ROESY, NOESY, or COLOC) [59,61] and a chiral anisotropic reagent (Figure 6) [62]. Yessotoxin (**50**) modulates the cellular calcium ion levels [63] and decreases the adenosine 3′,5′-cyclic monophospahete (cAMP) levels in human lymphocytes [64]. Likewise, it can induce apoptosis in the myoblast BC3H1 and L6 cell lines from mouse and rat, respectively [65].

Different synthetic methodologies have been reported by Oishi and coworkers to build the different ring systems in batch mode [66,67,68,69]. Thus, reductive etherification was applied as a key step for constructing the H-ring using Et_3_SiH/TMSOTf. Consequently, silyl groups were removed using TBAF and the FGHIJ ring unit (**52**) was afforded as a single diastereomer in a 74% yield for the two reaction steps (Figure 15) [70].

However, the scaled-up version of the reaction presented a problem, as the yield cut down to 19% and a hydroxy ketone (**53**) was obtained as a mixture of diastereomers (73% yield) (Figure 16) [71].

To avoid the formation of by-product (**53**), a microflow reactor was used for carrying out the reductive etherification (Figure 17). Therefore, two solutions were transmitted to the microflow reactor at −30 °C using a syringe pump at 25 mL/min. One of the solutions contained compound (**51**) (0.03 M) and Et_3_SiH (3 M) in dichloromethane, and the other TMSOTf (1 M) in the same solvent. Both solutions were mixed vigorously through a 50 μL volume reactor. Aqueous NaHCO_3_ at 0 ℃ was used to quench the resulting mixture which flowed out, and, after extraction with diethyl ether, the crude product was treated with TBAF in THF to give the desired FGHIJ ring unit (**52**). The new methodology was not affected by the reaction scale, because both solutions were continuously mixed when charged into the reactor and the yield remained around 86%, even if 1.5 g of the substrate is used [71,72].

### 2.6. Synthesis of Azaspiracids through Flow Photochemistry

Visible-light photocatalysis has emerged over the last years as a powerful synthetic tool in organic chemistry. This technology is based on the ability of certain photocatalysts to absorb light and promote the single electron transfer (SET) or energy transfer (ET) reactions of organic substances. It has been positioned as a selective, group-tolerant, and simple operational strategy to achieve unique reaction pathways. Thus, many synthetic applications have been reported in a variety of fields; for instance, it has been used in peptide functionalization, C(sp^3^)-C(sp^2^) cross-coupling reactions, or late-state functionalization processes [73]. The development of continuous flow photoreactors overcomes certain limitations of the technique as a more homogeneous irradiation is achieved, resulting in better selectivities and lower reaction times [74,75,76]. In the case of marine drugs, the synthesis of azaspiracids (AZA) has been reported using this strategy.

Azaspiracids are polyether marine toxins that are accumulated in various shellfish species and have caused human intoxication since 1995. In that year, at least eight people in the Netherlands became ill after eating mussels (*Mytilus edulis*) that have been cultivated at Killary Harbour, Ireland. Some years later, Yasumoto and coworkers reported that the cause of the incident was the presence of azaspiracids in the mussels *Mytilus edulis*. These authors isolated and determined the structure of azaspiracid-1 (AZA1) and azaspiracid-2 (AZA2) in 1998 and 1999, respectively [77,78]. In 2003, Nicolaou and coworkers achieved the synthesis of azaspiracid-1, but HPLC analysis revealed that it was an epimer of the natural product [79,80]. This discovery showed that the structure of AZA1 was incorrect and it was necessary to determine its real structure. After numerous degradation studies and the synthesis of multiple diastereomers, the correct structure of azaspiracid-1 (**57**) was determined (AZA1) in 2004 (Figure 7) [81,82]. Two years later, the revised structure of azaspiracid-2 (**58**) was also published (AZA2) [83].

During the following years, the total synthesis in batch mode of various azaspiracids such as (+)-azaspiracid-1 ((+)-**57**) [84,85], (−)-azaspiracid-1 ((−)-**57**) [86,87], and azaspiracid-2 (**58**) [83] was developed.

In 2009, it was reported that the primary producer of azaspiracid-1 (**57**, AZA1) and azaspiracid-2 (**58**, AZA2) was the dinoflagellate *Azadinium spinosum* [88]. This discovery enabled the possibility for us to obtain both azaspiracids in a culture of this micro-organism. Even though dinoflagellates are often considered sensitive to shear stress produced by small turbulences, *Azadinium spinosum* was able to grow in stirred photobioreactors. Thus, in 2012, Hess and coworkers described the production of azaspiracid-1 (**57**, AZA1) and azaspiracid-2 (**58,** AZA2) in a culture of *Azadinium spinosum* on a pilot scale by using flow photobioreactors [89].

The equipment was formed by two stirred photobioreactors in a series of 100 L each, followed by a harvesting tank of 300 L. Neon tubes provided light to one side of the photobiorector with a photon flux density at 200 μmol m^−2^ s^−1^ and a photoperiod of 16 h of light and 8 h of dark. After that, the feed reservoir and retentate were obtained, whose extraction gives AZA1 and −2 (Figure 8).

AZA extraction procedures were developed to optimize AZA recovery from the culture. Tangential flow filtration and continuous centrifugation were tested and both protocols were successfully applied. Continuous centrifugation is less time-consuming and can easily been scaled up compared to tangential flow filtration. However, the initial purchase cost is higher. Tangential flow filtration was the selected extraction methodology.

Once the feed reservoir and retentate were obtained, an extraction was needed to provide AZA1 (**57**) and AZA2 (**58**). The determination of volume solvent, the amount of resin, the tie of contact between the sample and the resin, and the method for AZA desorption from the resin were optimized to achieve a better extraction yield. When studying the effect of the dilution rate of *A. spinosum* and AZA production, a dilution rate of 0.25 day^−1^ yielded the highest volume-specific toxin production per day, which generates an AZA production of μg day^−1^ (19 μg day^−1^ L^−1^).

The higher concentration of AZAs in the matrix extract led to a four-step purification process with recoveries of 75% for AZA1 (**57**) and 70% of AZA2 (**58**), that is, a 1.5 increment compared with the extraction from shellfish. This study was carried out on six successive culture lots (1200 L harvested in total) and obtained 9.3 mg of AZA1 (**57**) and 2.2 mg of AZA2 (**58**) of >95% purity in 12 days (8 days of culture, 1 day of filtration, and 3 days of extractions), showing that the obtention of AZA1 and −2 in photobioreactors has been possible. However, only racemic mixtures are obtained with this methodology and batch protocols were needed to obtain pure enantiomers.

## 3. Conclusions

Novel technologies have been introduced during the last 20 years to improve certain limitations of chemical batch processes, such as low productivity, long reaction times, or lack of reproducibility. Between them, continuous flow chemistry has been positioned as a sustainable alternative in which the chemical process takes place from a greener chemical perspective. On the other hand, photochemical methodologies allow us to access unlocked chemical space and expand the synthetic toolbox.

Although these technologies have primarily been used for scale-up processes (continuous flow) or to achieve new molecular connectivities (photochemistry), they have also demonstrated how it is possible to carry out certain chemical reaction steps in the preparation of molecules from marine organisms. In some cases, both traditional batch and flow methodologies have been combined in the synthesis of different marine drugs to improve the efficiency of the reaction protocol. In others, telescope flow versions have been developed, reducing work-up and purification processes, which showcases the benefits of using this technique.

We anticipate that the investigation of flow chemistry in the development of marine drugs and their precursors will be developed extensively in the following years due to the recent tendency in other chemical fields to look for greener methodologies—for instance, adapting continuous flow photochemistry or even combining flow with other synthetic strategies (electrochemistry, artificial intelligence, etc.) and implementing these approaches in the development of more marine drug analogs.

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
