# Peer review of "Continuous Flow Chemistry: A Novel Technology for the Synthesis of Marine Drugs"

_marinedrugs, 2023, doi:10.3390/md21070402_

Round 1

Reviewer 1 Report

The manuscript marinedrugs-2482331 entitled "Continuous flow-chemistry and photochemistry: novel technologies for the synthesis of marine drugs submitted as review article describes the advantages of innovative enabling technologies as flowchem and photochem for the preparation of biologically-active compounds of marine origin. After a brief introduction on the technologies, the Authors discussed the application of both technologies providing interesting examples. 

Specific comments:

1. The introduction section is generally well-written. However, some crucial reviews in the field are missing. E.g.: a) Chem. Soc. Rev. 2017, 117, 11796, which is one of the more comprehensive for what concern the general principles and advantages of flowchem; b) J. Med. Chem. 2020, 63, 6624 which relies with the automation of flow chemistry platforms in medchem settings; c) Chem. Sci., 2023,14, 4230 and Chem. Rev. 2022, 122, 2752 which provides nice examples for the combination of flow chemistry with photochemistry and electrochemistry; d) J. Flow Chem. 2017, 7, 65  that describes the application of flow technology for taming forbidden chemicals; e) Adv. Synth. Catal. 2023, 365, 1-26 that describes the application of flow chem, photochem and biocatalysis for the preparation of natural and synthetic steroids. I would suggest to add these very recent reviews where appropriate in the introduction paragraph.

2. Paragraph 2.1 Continuous flow chem. For this paragraph, I have really appreciated the efforts made by the Authors to compare batch and flow synthesis. However, some crucial points should be better addressed in my opinion:

a. Scheme uniformity. All flow set-up should report the main experimental parameters: reactor type and volume, temperature, residence time, flow rate, reagents concentration and solvent. I would suggest to uniform the schemes.

b. Comparison batch vs flow: beyond the overall yield, additional parameters should be taken into account to have a better comparison: e.g. productivity data and space-time-yield data are missing.

c. While the Authors accurately described the batch process (maybe too much in some part e.g. for compound 15) no specific information have been provided in terms of specific equipment used for photo and flow chem reactions (e.g. type of pumps, reactors, lamp, and so on).

3. P9 L216: Please check the correct flow rate. It should be 83 microliter per minute.

Based on these consideration, I cannot accept this perspective in the present form but I would be happy to revaluate a revised version based on my comments/suggestions.

English style is usually good

Reviewer 2 Report

This review collects a series of interesting articles on the applications of marine drugs synthesis and analogs using flow reactor technology. In this regard, a comprehensive introduction has been provided regarding the advantages associated with the use of this technology. The comparison between traditional batch synthesis and flow synthesis is emphasized in some examples. However, the productivity for minute or hour data is missing in almost all the referenced articles. Productivity for hour is important point when discussing continuous flow approach. I would suggest extracting this data if it is present in the original article. Furthermore, I would recommend revising figures 5, 6, 7, 8, 9, 10, 11, 12, 13, 14, 15, 16, 17, 18, 19, and 20, as there are alignment errors in the characters as well as in some structures. Regarding visible light-induced photocatalysis technology, it appears that this part of the review is underdeveloped. There are two examples, one describing both flow chemistry and photobiocatalysis technologies, the other one a synthesis of cyclopeptides using UV vis irradiation. Unfortunately, both technologies, flow and photo, are included in the title of the review but are not well-balanced. I would suggest incorporating the photochemistry part as a subsection of flow chemistry and removing the synthesis of galaxamide. If the authors could include additional examples of photochemistry, then the proposed title and subdivision of the review might be appropriate.

After addressing the suggestions and implementing the proposed changes, the review could be considered suitable for publication in Marine Drugs.

Reviewer 3 Report

The authors of the review have put together a nice collection of examples about the use of flow chemistry to improve the synthesis of different marine drug examples. The review is very comprehensive as they covered different drug examples, from nucleosides to complex alkaloids. They also covered the use of different technologies as well such as biocatalysis, photochemistry and photobiocatalysis. For all this reasons I found the review appropriate for publishing in this journal in its current form.

Round 2

Reviewer 1 Report

I am fully satisfied of the revisions. All my comments were taken into account. The manuscript reaches the quality level for publication.